# Analysis of conflict of interest policies among organizations producing clinical practice guidelines

J. Henry Brems[1]*, Andrea E. Davis[1], Ellen Wright Clayton[2,3]

1 Department of Medicine, Vanderbilt University Medical Center, Nashville, Tennessee, United States of America, 2 Center for Biomedical Ethics and Society, Vanderbilt University Medical Center, Nashville, Tennessee, United States of America, 3 Law School, Vanderbilt University, Nashville, Tennessee, United States of America

* john.h.brems@vumc.org

## Abstract

### Background

Conflicts of interest (COI) jeopardize the validity of Clinical Practice Guidelines (CPGs). When the Institute of Medicine promulgated COI policies in 2011, few organizations met these requirements, but it is unknown if organizations have improved their policies since that time. We sought to evaluate current adherence to IOM standards of COI policies.

### Methods and findings

We conducted a retrospective document review of COI policies and CPGs from organizations that published five or more CPGs between January 1, 2018 and December 31, 2019. Organizations were identified via CPG databases. COI policies were obtained from an internet search. We collected data on i) the number of organizations that have COI policies specific to CPG development, ii) the number of policies meeting each IOM standard and iii) the number of IOM standards met by each policy. COI disclosures from five CPGs of each organization were assessed for adherence to IOM standards. Among the 46 organizations that published 5 or more CPGs, 36 (78%) had a COI policy. Standard 2.2b (requiring divestment of financial COI) was met least frequently, by 2 of 36 (6%) organizations. Standard 2.1 (requiring disclosure of COI) was met most frequently, by 33 of 36 (92%) organizations. A total of 31 of 36 (86%) organizations met 4 or fewer of the 7 IOM standards. Among the 16 organizations limiting COI to a minority of the CPG panel (standard 2.4c) and the 15 organizations prohibiting COI among chairs or co-chairs (standard 2.4d), 12 (75%) and 10 (67%) organizations violated the respective standard in at least one CPG. The main limitations of our study are the exclusion of organizations producing fewer CPGs and ability to assess only publicly available policies.

### Conclusion

Among organizations producing CPGs, COI policies frequently do not meet IOM standards, and organizations often violate their own policies. These shortcomings may undermine the

**Data Availability Statement:** All relevant data are within the paper and its Supporting information files.

**Funding:** The authors received no specific funding for this work.

**Competing interests:** The authors have declared that no competing interests exist.

public trust in and thus the utility of CPGs. CPG-producing organizations should improve their COI policies and their strategies to manage COI to increase the trustworthiness of CPGs.

## Introduction

Clinical Practice Guidelines (CPGs) provide important recommendations for healthcare providers on the prevention, screening, diagnosis, and treatment of a wide variety of diseases [1]. As defined by the Institute of Medicine (IOM) (now the National Academy of Medicine) in its 2011 report, CPGs are documents that formulate "recommendations intended to optimize patient care" based on a systematic review of the available evidence and an assessment of the risks and benefits [2]. Even when based on high-quality evidence, the CPG development process requires individuals to make judgments, which inherently introduces the potential for bias. In some cases, CPG developers do not have access to robust high-quality evidence [3], requiring even greater reliance on panel members' individual interpretations. Thus, CPGs can be susceptible to individuals' conflicts of interest (COI).

The IOM's earlier 2009 report addressing COIs defined them as "circumstances that create a risk that professional judgments or actions regarding a primary interest will be unduly influenced by a secondary interest" [4]. Specifically within CPG development, COI between the primary goal of creating optimal guidance and a committee member's other secondary interests may influence CPG development at any stage including interpretation of the evidence, drafting of recommendations, or voting on recommendations; and COI can take various forms [5]. Financial COI may occur when an individual has a secondary interest that could result in financial gain, and it can be further subcategorized into commercial or non-commercial COI depending on whether the source of financial gain comes from a commercial source (e.g. consulting, board membership, consultancy fees, industry-sponsored research) or a non-commercial source (e.g. government research grants) [2]. In addition, nonfinancial COIs may occur and negatively impact CPG development, such as intellectual COI when an individual has an interest in promoting their own research or academic activity [2].

While more data on the effect of COI on CPG recommendations are needed [6], limited research suggests that COI can have a significant impact. In particular, studies have shown that endorsement of a certain drug is more common when authors have an associated financial COI [7,8]. As such, conflicts of interest (COI) threaten the validity of CPGs [4].

To address the challenge of COI for development of CPGs and building on its earlier report in 2009 focused on COI throughout medicine, the IOM published a report in 2011 on trustworthy guidelines. The report delineated multiple standards that should be met when producing CPGs, including a requirement that they "be based on an explicit and transparent process that minimizes distortions, biases, and conflicts of interest" [2]. Despite the increased attention that the IOM report has brought to COI within CPG development, significant concern remains regarding COI within CPG panels and the policies for managing COI within the development process [9]. More recent studies have shown that COI among guideline authors remain prevalent in most specialties, including but not limited to cardiology [10,11], oncology [12], gastroenterology [13], endocrinology [14], infectious disease [11], psychiatry [15], dermatology [16], otolaryngology [17], and urology [18]. Moreover, the prevalence of financial COI has been reported to be particularly high in CPGs related to high revenue medications

[19], and authors' conflicts may frequently be related to the drugs they are recommending in the CPG [20].

A study examining CPGs promulgated in the two years prior to the IOM CPG report found that few organizations had COI policies in place for CPG development that met the 2011 IOM standards [21]. Since then, many guideline-producing organizations have updated their COI policies and made explicit reference to the IOM standards when doing so [22–25]. Nonetheless, recent, smaller studies suggest adherence to the IOM standards is still inadequate [11].

To our knowledge, adherence of COI policies to IOM standards has not been systematically evaluated since immediately after the IOM CPG report [21]. Thus, it is unknown whether CPG-producing organizations have since created, updated, or adhered to policies that meet IOM standards to ensure the development of trustworthy CPGs. We conducted an analysis of COI policies and their implementation among organizations producing CPGs to assess the current level of adherence of CPG-producing organizations to IOM standards.

## Materials and methods

### Sample selection

We sought to replicate the strategy previously used by Norris and colleagues [21] to assess how organizations have responded since the publication of the IOM's standards. As the National Guidelines Clearinghouse database of CPGs is no longer available due to a loss of funding [26], and no current single database is similarly comprehensive to our knowledge, we searched three databases–Emergency Care Research Institute, Guidelines International Network, and Medscape—for all CPGs published in the two-year period from January 1, 2018 to December 31, 2019. We additionally searched the website of every organization identified from the three databases to ensure all CPGs from the eligible period were obtained. All publications were then reviewed to ensure they met the IOM definition of a CPG [2]. Thus, we considered only documents published in English that included (i) a systematic review, (ii) an assessment of benefit and harms, and (iii) distinct recommendations. We included guideline 'updates' as distinct CPGs if they similarly met all three criteria.

All organizations that published five or more CPGs in the eligible time frame were included in the study—criteria that have been used previously in an evaluation of COI policies prior to publication of IOM standards [21], thus allowing for a comparison to evaluate the response of CPG-producing organization in the interim. Additionally, this population of organizations was felt most likely to have developed COI policies since they are most actively engaged in the production of CPGs and thus perhaps most likely to have created standard processes for CPG production.

### COI policy search

A two-step strategy was employed to find COI policies for each organization. One author (JHB) first searched each organization's website. In addition, that author conducted an internet search using www.Google.com with the string: (organization name) and ("conflict of interest policy," "COI policy," "guideline COI," guideline conflict of interest," "guideline development," OR "guideline methodology"). All potentially relevant documents were collected for review. We did not contact organizations directly for COI policies as we wanted to include only publicly available information. Transparency is important in promoting trustworthiness, and the IOM recommended that information regarding CPG development processes be publicly accessible in its 2011 report [2].

## COI policy assessment

Two authors (JHB, AED) independently reviewed all COI documents obtained in the search. Similar to Norris and colleagues [21], we excluded those policies that did not address COI specific to CPG-development.

For all policies specific to CPG-development, the two authors used a standardized form to abstract data from the policies regarding disclosure and management of COI. The two authors then independently judged which of the seven IOM standards listed in Table 1 were met by each organization's policy(s). These standards were chosen from the report as they are the ones that can be evaluated within a policy and have been used previously [21]. Although the IOM is an American body, the IOM standards are still useful to evaluate documents produced by international organizations because clinicians should be wary of any CPG, regardless of the region in which it was developed, that does not meet these standards governing COI. As such, the IOM standards have been used in previous studies of non-US organizations [14,21]. Moreover, organizations outside of the U.S. have made explicit reference to the IOM standards [27,28] in relation their own COI policies.

Although a similar approach to collect data from COI policies using a standardized form has been used previously [29], the current form was designed specifically for this study to systematically obtain information deemed necessary to make a judgment on adherence to IOM standards or to otherwise evaluate relevant policy characteristics as reported by prior studies [21]. The full form, including which parts were used to make a judgment on IOM standards, is available in S1 Appendix. Data from both authors were then reviewed, and any differences were reconciled via in-person consultation. Cohen's kappa was calculated using Microsoft Excel to assess inter-rater reliability. It was calculated using the authors' independent judgments of whether a standard was met or not met for all standards and for all organizations.

## Assessment of COI disclosures

We additionally evaluated the author COI disclosures of five CPGs from each organization for adherence to IOM standards and to the organization's policy, if available. For organizations with only five CPGs published in the study period, all CPGs were reviewed for disclosure. For

**Table 1. Institute of Medicine Standards.**

| Institute of Medicine Standards[a] | |
|---|---|
| 2.1 | Prior to selection of the Guideline Development Group (GDG), individuals being considered for membership should declare all interests and activities potentially resulting in COI with development group activity, by written disclosure to those convening the GDG. Disclosure should reflect all current and planned commercial (including services from which a clinician derives a substantial proportion of income), non-commercial, intellectual, institutional, and patient/public activities |
| 2.2a | All COI of each GDG member should be reported and discussed by the prospective development group prior to the onset of their work |
| 2.2b | Each panel member should explain how their COI could influence the CPG development process or specific recommendations |
| 2.3 | Members of the GDG should divest themselves of financial investments they or their family members have in, and not participate in marketing activities or advisory boards of, entities whose interests could be affected by CPG recommendations |
| 2.4a | Whenever possible GDG members should not have COI |
| 2.4c | Members with COIs should represent not more than a minority of the GDG |
| 2.4d | The chair or co-chairs should not be a person(s) with COI |

[a]Standards are taken verbatim from 2011 IOM report. (2) Although more standards exist in the report, these are the seven that pertain to COI and are able to be evaluated within COI policies. [21].

organizations with more than five CPGs, only five were chosen for evaluation at random via a random number generator at www.random.org. One author (JHB) then reviewed each CPG and its corresponding appendices for the presence of author COI disclosures. Organization websites were searched for COI disclosures only if this was explicitly referenced by the guideline.

If a CPG only included a statement to the extent that "the authors have no COI to disclose," this was judged as the presence of a disclosure. CPGs were judged to have no disclosure only if there was a complete absence of any statement referring to COI disclosure or related terms.

For CPGs that provided information on "relevant" COI in addition to all COI disclosed by authors, we only considered the section of "all disclosed" COI, as the IOM standards do not make a distinction between the two. Further, the determination of "relevant'" COI was considered specific to a single organization's judgment and thus would impair comparability to other organizations' disclosures.

Data were then collected from the disclosure section on the number of authors with a declared COI, the number of chairs or co/vice-chair with a declared COI, and whether the COI was a commercial COI. Commercial COI were considered any financial interests with industry including consulting, board membership for which compensation is received, service as an expert witness, industry-sponsored research, patents, royalties, stock ownership, or other financial interests [2].

Due to the nature of the information available, we could only assess disclosures for adherence to the limitation of COI to a minority of the CPG panel (i.e., whether fewer than 50% of panel members disclosed a COI, as per standard 2.4c) and to the prohibition of COI among chairs and co-chairs (standard 2.4d). For all organizations whose COI policies met one or both of those standards, we assessed whether author disclosures within their CPGs similarly met the standards.

Descriptive statistics were calculated for all data using Microsoft Excel.

## Results

### Organization and policy characteristics

We identified 907 CPGs published between January 1, 2018 and December 31, 2019. A total of 46 organizations produced five or more CPGs in that period, and these organizations accounted for 702 (77%) of the CPGs identified. The 46 organizations included 34 (74%) professional societies, 9 (20%) government organizations, 2 (4%) health care providers, and 1 (4%) medical journal. A total of 25 (54%) organizations were from the U.S., while 17 (37%) were from Europe, 3 (6%) from Canada, 1 (2%) from Asia. No organization were identified from Africa or South America.

Among the 46 organizations, 36 (78%) had a publicly available policy that addressed COI specific to production of CPGs. Of these 36 policies, 29 (81%) indicated when they were published. A total of 17 (47%) policies were published after January 1, 2018, and 25 (69%) policies were published after January 1, 2015. Further characteristics of all organizations and of those with COI policies are shown in Table 2. A full list of organizations and corresponding policy characteristics can be found in S1 Table in S1 Appendix.

### Disclosure

All 36 policies required CPG panel members to provide COI disclosures. Almost all (92%) policies specified that disclosures must occur prior to the authors' starting work on the guideline. A total of 19 (53%) policies also required ongoing disclosures throughout the authors' time

**Table 2. Organization demographics.**

| Type of Organization | No. (%) Organizations (N = 46) | No. (%) Organizations with a COI policy (N = 36) |
|---|---|---|
| US Professional | 20 (43) | 15 (42) |
| US Government | 3 (7) | 3 (8) |
| Canada Professional | 1 (2) | 1 (3) |
| Canada Government | 2 (4) | 2 (6) |
| Europe Professional | 13 (28) | 10 (28) |
| Europe Government | 3 (7) | 3 (8) |
| Asia Professional | 1 (2) | 0 (0) |
| Asia Government | 0 (0) | 0 (0) |
| Other[a] | 3 (7) | 2 (6) |

[a]Other organizations include 1 European medical journal, 1 U.S. health care provider, and 1 U.S. non-governmental organization. A full list of organizations is available in S1 Appendix.

working on the guideline. Only 2 (6%) policies did not specify a timeline for which COI disclosure need to be reported.

A total of 28 (78%) policies specifically required disclosure of financial COI. A minority of 15 (42%) policies required disclosure of intellectual COI, and 6 (17%) policies required "non-financial" COI.

Most organizations (28, 78%) did not specify a monetary threshold required for disclosure. A total of 6 (17%) policies explicitly required the disclosure of all financial COI (i.e., a threshold of $0). Only 2 (6%) organizations specified a threshold for disclosure that was greater than $0. Similarly, a total of 5 (14%) organizations specified a monetary threshold for which financial COI was disqualifying, which ranged from $1,000 to $50,0000. Conversely, 31 (86%) policies did not include a threshold for which COI was disqualifying.

## Management of COI

Five (14%) organizations required CPG panel members to divest themselves of financial COI prior to starting work on the guideline, while 30 (83%) organizations had no such requirement. Additionally, 1 (3%) organization required divestment in only some circumstances.

Strategies for managing COI among CPG panel members varied. A total of 12 (33%) policies required conflicted members to be recused from working on or developing the section relevant to their COI. A similar number of 14 (39%) policies required members with a COI to be recused from voting on the recommendations relevant to their COI. Seventeen (47%) organizations did not list a strategy for managing members' COI.

Regarding the composition of CPG panels, a total of 16 (44%) policies required that a majority of panel members must be free of COI (IOM standard 2.4c), while the other 56% did not. Fifteen (42%) organizations required all chairs or co-chairs to be free of COI, while 14 (39%) organizations had no such requirement. A total of 7 (19%) organizations allowed one chair or co-chair to have a COI if there were other chairs or co-chairs who do not have a COI.

## Compliance with Institute of Medicine Standards

Among the 36 organizations with policies, nearly all (34, 94%) met at least one standard, although 2 (6%) did not meet any standard, and only 1 (3%) met all seven standards. Only 5 (14%) organizations met five or more IOM standards compared to 31 (86%) organizations that met four or fewer. Fig 1 shows the number of standards met by organizations and demonstrates how this varied by the organizations' region.

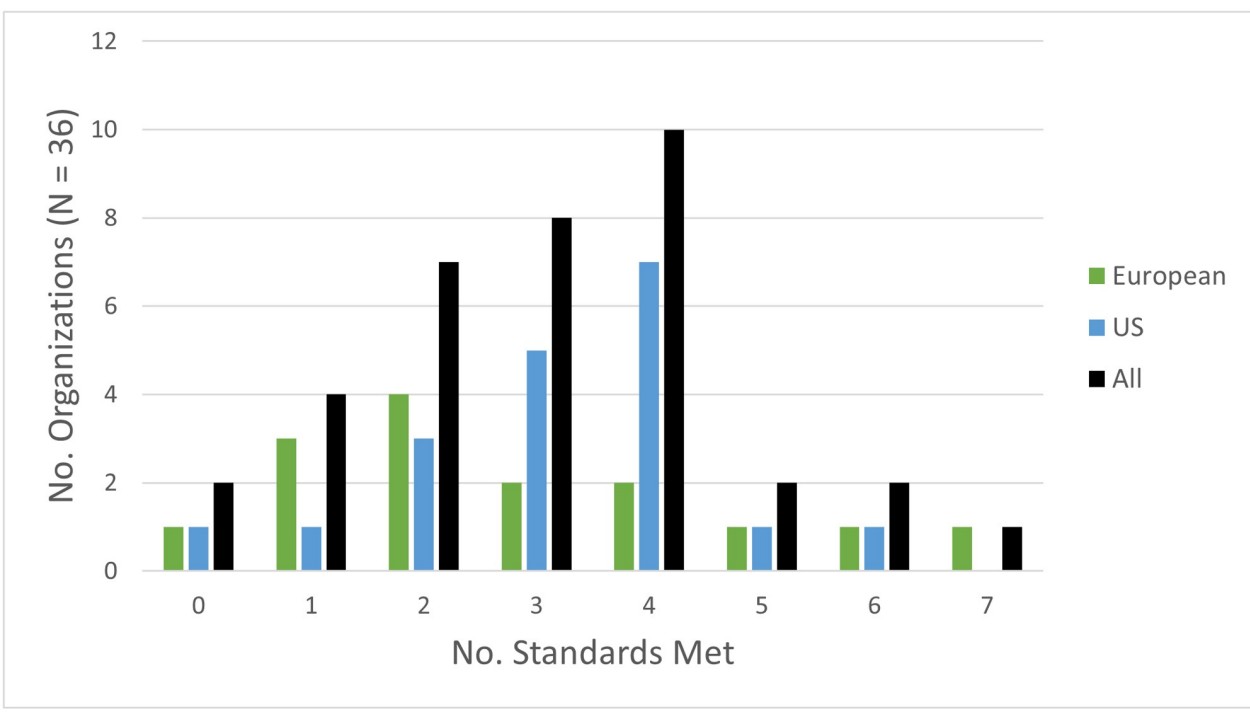

**Fig 1. Number of IOM standards by organizations.** The number of organizations (y-axis) that meet a total of *x* standards (x-axis) are shown. For each *x* number of standards met, results are stratified per the legend to show the number of organizations from Europe, the US, or everywhere (i.e. All organizations) meeting *x* standards.

The most frequently met standard was 2.1 regarding disclosure of COI, which was met by 33 (92%) organizations. Standard 2.2b regarding explanation of one's COI was met the least often, as only 2 (6%) organizations met this standard. Table 3 demonstrates the frequency with which each specific IOM standard was met. S1 Table in S1 Appendix displays the number of

**Table 3. Number of organizations with COI policies meeting each IOM standard, stratified by region and type of organization.**

| | IOM Standards | | | | | | |
|---|---|---|---|---|---|---|---|
| | 2.1 (disclosure of COI) | 2.2a (report of COI) | 2.2b (explanation of COI) | 2.3 (divestment of financial COI) | 2.4a (should not have COI) | 2.4c (limitation on panel) | 2.4d (limitation on chairs) |
| **No. (%) met by Organization** | | | | | | | |
| Region | | | | | | | |
| *US (N = 19)* | 18 (95) | 17 (89) | 1 (5) | 3 (16) | 6 (32) | 10 (53) | 6 (32) |
| *Europe (N = 14)* | 12 (86) | 8 (57) | 1 (7) | 3 (21) | 5 (35) | 5 (35) | 7 (50) |
| *Canada (N = 3)* | 3 (100) | 3 (100) | 0 (0) | 0 (0) | 0 (0) | 1 (33) | 2 (67) |
| Type of Organization | | | | | | | |
| *Professional (N = 26)* | 23 (88) | 19 (73) | 2 (8) | 5 (19) | 8 (31) | 13 (50) | 10 (38) |
| *Government (N = 8)* | 8 (100) | 7 (88) | 0 (0) | 0 (0) | 2 (25) | 2 (25) | 4 (50) |
| *Other (N = 2)* [a] | 2 (100) | 2 (100) | 0 (0) | 1 (50) | 1 (50) | 1 (50) | 1 (50) |
| All (N = 36) | 33 (92) | 28 (78) | 2 (6) | 6 (17) | 11 (31) | 16 (44) | 15 (42) |

[a]Other organizations included 1 European medical journal and 1 U.S. non-governmental organization.

**Table 4. Violations of IOM standards 2.4c and 2.4d among organizations with COI policies that meet the respective standard, stratified by region and type or organization.**

| | IOM Standard 2.4c (panel members with COI restricted to a minority) | IOM Standard 2.4d (Chairs and co-chairs should not have COI) |
|---|---|---|
| **No. (%) Organizations with 0 CPGs violating standard[a]** | | |
| All organizations | 4 (25) | 4 (27) |
| *US organizations* | 1 (10) | 1 (17) |
| *European organizations* | 3 (60) | 2 (29) |
| *Canadian organization* | 0 (0%) | 1 (50%) |
| *Professional organizations* | 2 (15) | 0 (0) |
| *Government organizations* | 1 (50) | 3 (75) |
| **No. (%) Organizations with 1 or more CPG(s) violating standard[a]** | | |
| All organizations | 12 (75) | 10 (67) |
| *US organizations* | 9 (90) | 5 (83) |
| *European organizations* | 2 (40) | 4 (57) |
| *Canadian organizations* | 1 (100%) | 1 (50%) |
| *Professional organizations* | 11 (85) | 9 (90) |
| *Government organizations* | 1 (50) | 1 (25) |

[a]Percentages are calculated from the number of organizations with policies meeting the respective standard, as found for each subgroup in Table 3.

standards met by each individual organization. Cohen's kappa for interrater reliability between both reviewers was 0.72 (95% Confidence interval; 0.64–0.81).

## Assessment of disclosures

We evaluated the COI disclosures from 230 CPGs of all 46 organizations included in the study. A total of 205 (89%) CPGs included COI disclosure for the CPG authors. Of the 46 organizations, 35 (76%) provided COI disclosures for all CPGs that were searched. A total of 3 (7%) organizations did not provide disclosure for any of five CPGs, and an additional 8 (17%) organizations published at least one CPG without COI disclosure.

Of the 35 organizations that included disclosures in all CPGs, 14 (40%) complied with standard 2.4c limiting the number of panel members with COI in any CPG. Of the 28 organizations that included disclosures in all CPGs and specified the chair or co-chair(s), only 6 (21%) met standard 2.4d prohibiting COI among chairs or co-chairs in all CPGs.

Among the 16 organizations that had a policy that limited COI among panel members (standard 2.4c), only 4 (25%) organizations met this standard in each CPG, whereas 12 (75%) organizations violated this standard in at least one CPG. Similarly, among the 15 organizations whose policy prohibited COI among chairs or co-chairs (standard 2.4d), a total of 4 (27%) organizations met this standard in each CPG, and 10 (67%) organizations violated this standard in at least one CPG. We identified a single organization whose policy met standard 2.4d but who did not specify the chair or co-chairs in their CPGs. These results are detailed in Table 4 and stratified by the region and type of organization.

## Discussion

We found that among organizations that have produced five or more CPGs in a two-year period, a significant portion (22%) did not have a COI policy, and of those that did, few met all

or nearly all IOM standards. Although our findings show better adherence of COI policies to the IOM standards compared to a prior assessment [21], these results demonstrate a need for further improvement in COI policies managing guideline development. Our findings raise several points worth highlighting.

First, COI policies in practice fail to meet the IOM's standards for the trustworthy development of CPGs. Although the COI policies in our study generally required disclosure (IOM standards 2.1, 2.2a), very few (6%) required the members to explain how their COI could affect the guideline development process (IOM standard 2.2b). Strategies to manage COI were frequently lacking with only 44% and 42% requiring limitations on the number of panel members or chairs with COI, respectively. Although disclosure is a critical first step, disclosure alone paradoxically may actually lead experts to provide more biased advice [30,31]. Better strategies to manage these disclosures are necessary to limit bias, which can cause potential harm to patients.

Although adherence to IOM standards in our study was imperfect, there has nonetheless been notable improvement since the publication of the IOM standards. In a prior similar study by Norris et al, only 48% of organizations had COI policies, and of those policies, 53% did not meet a single IOM standard [21]. In comparison, our study found that 78% of organizations have COI policies, and only 2 policies (6%) failed to meet any IOM standards. Clearly, organizations have taken some steps.

Even so, organizations could further improve their management of COI within the CPG development process by increasing transparency. As we found that 22% of organizations did not have publicly available COI policies and that 11% of CPGs did not include author COI disclosures, organizations need to commit to greater transparency, a policy that both physicians and non-physicians are increasingly adopting regarding research [32]. The establishment of the Open Payments Program has helped improve transparency [33], and by making use of it, multiple studies have demonstrated that COI disclosures among CPG authors are often incomplete [16,17,34].

Additionally, transparency at the organizational level is critical. While it was not evaluated in our study, IOM standard 1.1 from the 2011 report stated that the "processes by which a CPG is developed and funded should be detailed explicitly and publicly accessible"[2]. Industry funding of a CPG represents a particularly concerning COI, as it may bias the CPG toward recommending a product from the funder [4]. If present, organizational COI represents another threat to the trustworthiness of CPGs and thus only accentuates the need for organizations to meet the IOM standards and improve their own transparency.

Current options for external policing of transparency and adherence to IOM standards are unfortunately limited. The National Guidelines Clearinghouse, which was hosted by the Agency for Healthcare Research and Quality (AHRQ), was inactivated in 2018 [26], despite the identification of this database in the 2011 IOM report as important tool for disseminating trustworthy guideline [2]. Alternatively, journals could enact more stringent standards for publication of CPGs [26]. For example, they could refuse to publish CPGs that do not meet IOM standards. However, CPGs are frequently published in professional societies' own journals, which may have strong incentives to publish CPGs as they are often among the most-cited articles in the medical literature, which though highly desirable can nonetheless represent yet another COI [35]. Reestablishment of the AHRQ or similar, centralized database by a government or independent body could promote adherence to IOM standards either by refusal to include CPGs that do not meet some minimum number of standards or by notifying database users of which standards are met by the given CPG.

For now, the onus remains on CPG developers to commit to increased transparency and adherence to standards themselves. Among the organizations whose COI policy limits COI to

a minority of panel members (standard 2.4c) or prohibits COI among chairs or co-chairs (standard 2.4d), only 25% and 27% respectively adhered to those standards in all of their CPGs examined in our study. While publicly available COI policies are meaningful and necessary [36], a violation of an organization's own policy significantly undercuts the value of these policies. Further, it threatens the trustworthiness of all CPGs published by that organization.

Although it is possible that our study may have over-identified violations by including "potential" COI or of COI disclosures that were reported by authors but not deemed relevant by the organization, this is unlikely for multiple reasons. First, "potential COI" is a misleading term—a COI exists or it does not [37]. If a COI is reported, it should be assumed to accurately reflect a COI. Second, we included only commercial COI in our assessment of COI disclosures; however, organizations often collect other forms of COI such as intellectual COI [38], which may be nearly as common as financial COI [39] and also threaten the quality of CPGs [8]. Third, as noted before, CPG authors often underreport their COIs [16,17,34]. We considered a statement that "the authors have no COI to disclose" as the presence of a disclosure of no COI for all authors within a CPG. However, this was a limitation of our study because a single, generalized statement may indicate a greater degree of underreporting compared to CPGs that disclose the COI (or lack thereof) for each individual author.

Although our study is focused on the extent to which published COI policies adhere to IOM standards and so does not provide evidence of how bias may affect CPG recommendations, limited prior data suggest that COI can undermine their objectivity. Thus, recommendation of a drug may be more likely when authors have a COI with the pharmaceutical company producing that drug [7]. Major concerns that COI could have inappropriately affected the ACC/AHA CPG on hypertension recently led a major primary care professional society not to endorse them [10]. More research to assess the impact of COI on the validity of CPGs is clearly warranted.

Our results should be interpreted in the context of multiple limitations. First, our sample may not be representative of all CPG-producing organizations since we only included organizations that published five or more CPGs in a two-year period and only evaluated organizations with publicly available COI policies However, organizations that produce the most CPGs should be more likely to have developed policies since they are the most engaged in CPG development. Moreover, the subset of these organizations that publish their policies online may be more likely to be committed to transparency and to promoting public trust, and thus more likely to meet more IOM standards. While our results do not contain any organizations from Africa or South America, this finding may reflect a limitation of our sampling. Because we were unable to use the AHRQ database and because we only evaluated CPGs in English, it is possible we did not identify CPG-producing organization from other regions. However, it also possible that organizations from those regions produce fewer CPGs, and thus did not meet our inclusion criteria of having published 5 CPGs in the two-year window. Nonetheless, our findings may not be generalizable to organizations outside of the U.S., Canada, or Europe given our sampling.

Second, the IOM standards were not created as an objective tool for measurement. Thus, our findings cannot necessarily be interpreted as an entirely comprehensive review of COI management by organizations. Nonetheless, organizations that meet more standards may be reasonably expected to manage COI more effectively than organizations meeting fewer. At the least, meeting publicly available standards promotes trustworthiness of CPGs. Third, assessing the language in policy documents for adherence to standards requires interpretation, which is an important source of bias, albeit one we sought to address by having two coders, and may limit comparison to prior data. Reassuringly, our interrater reliability score indicates substantial agreement. Finally, only a single author reviewed COI disclosure within CPGs and could

have missed some disclosures. If so, our data would underestimate the frequency with which organization violate IOM standards.

## Conclusions

In conclusion, among organizations that produce CPGs, COI policies frequently do not meet IOM standards, a failing that threatens to undercut public confidence in these processes. Despite improvements since the publication of the 2011 IOM standards, organizations producing CPGs need to improve transparency, management of existing COI, and adherence to their own policies.

## Supporting information

**S1 Appendix.**
(DOCX)

## Author Contributions

**Conceptualization:** J. Henry Brems, Ellen Wright Clayton.

**Data curation:** J. Henry Brems, Andrea E. Davis.

**Formal analysis:** J. Henry Brems, Andrea E. Davis.

**Investigation:** J. Henry Brems, Andrea E. Davis.

**Methodology:** J. Henry Brems, Ellen Wright Clayton.

**Project administration:** J. Henry Brems, Ellen Wright Clayton.

**Supervision:** Ellen Wright Clayton.

**Visualization:** J. Henry Brems.

**Writing – original draft:** J. Henry Brems.

**Writing – review & editing:** J. Henry Brems, Andrea E. Davis, Ellen Wright Clayton.

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
