## [Decision Letter · Decision Letter 0]

19 Feb 2021

PONE-D-21-02302

Analysis of conflict of interest policies among organizations producing clinical practice guidelines

PLOS ONE

Dear Dr. Brems,

Thank you for submitting your manuscript to PLOS ONE. After careful consideration, we feel that it has merit but does not fully meet PLOS ONE’s publication criteria as it currently stands. Therefore, we invite you to submit a revised version of the manuscript that addresses the points raised during the review process.

The reviewers were highly positive about this paper and make a number of clarifying comments. Please address these points, with particular attention to the discrepancies noted among the Tables and in the generalisability of the IOM standards to non-United States guideline developers.

We look forward to receiving your revised manuscript.

Kind regards,

Quinn Grundy, PhD, RN

Academic Editor

PLOS ONE

Journal Requirements:

Reviewers' comments:

Reviewer's Responses to Questions

**Comments to the Author**

1. Is the manuscript technically sound, and do the data support the conclusions?

Reviewer #1: Yes

Reviewer #2: Yes

Reviewer #3: Yes

Reviewer #4: Yes

2. Has the statistical analysis been performed appropriately and rigorously? 

Reviewer #1: Yes

Reviewer #2: Yes

Reviewer #3: Yes

Reviewer #4: Yes

3. Have the authors made all data underlying the findings in their manuscript fully available?

Reviewer #1: Yes

Reviewer #2: Yes

Reviewer #3: Yes

Reviewer #4: No

4. Is the manuscript presented in an intelligible fashion and written in standard English?

Reviewer #1: Yes

Reviewer #2: Yes

Reviewer #3: Yes

Reviewer #4: Yes

5. Review Comments to the Author

Reviewer #1: This paper looks at whether organizations have policies about COI in CPGs that meet the IOM standards and whether individual CPGs published by those organizations meet the standards set out in the policies.

1. Why were the authors expecting that the European and Canadian organizations should have policies that met the criteria of the IOM, an American body?

2. Previous research has shown that CPGs developed by not-for-profit bodies and government agencies have fewer committee members with COI than CPGs developed by medical specialty societies and professional organizations (BMJ 2011;343:d5621). Was there a difference between these different types of organizations in this study?

3. In the Discussion, the authors should note the limitations in the IOM standards, specifically that the IOM standards apply to individual guideline committee members and not to the organizations themselves.

4. In the Introduction the authors should specify how they are defining COI.

5. In the Methods the authors should state whether there were language restrictions in their search.

6. Table 2 lists 3 Canadian organizations and 2 "other" organizations. How are these categorized in Table 3 in terms of "region"?

7. Where are the Canadian organizations in Table 4?

Reviewer #2: The authors analyzed COI policies from major CPG producers in order to find out whether the COI standards outlined by the IOM committee, Clinical Practice Guidelines We Can Trust, published in 2011, are being met. Those standards seem to have been widely accepted and have not been challenged; however, it is not known to what degree the standards are being adhered to in current years. There have been no recent evaluations and, to this reviewer’s perspective, the COI in medical studies and CPG’s may have increased, as evidenced by the recent controversy over the Excel trial. Further, the Covid-19 era has brought forth a great many “experts” whose COI is veiled. Thus, this study is timely and important and will provide the public with solid information to the degree the study is well conducted and is able to overcome the limitations imposed by its retrospective nature.

In my view, the study has succeeded. Its strengths are:

- The study attempts to compare current results with the earlier study of Norris et al. It is not strictly comparable but provides some baseline for comparison.

- The study goes beyond disclosures into the deeper, and really more important coi management issues which include: committee composition, coi by chairs, explanation of how coi could affect the results, non-recusal for members with coi, etc. These issues are critical because, as is now appreciated, disclosures alone are not adequate.

- They also note how the organizations violated their own policies, revealing the great pressure to develop guidelines that fit with the thought leaders in the organization rather than assure that the evidence determines the recommendations.

- The kappas for reliability are high.

The weaknesses include:

- Sampling: They identified the cpgs from several sources because the clearing house has been dissolved. These organizations might not be entirely representative, however, there is a very strong argument that the sampling is adequate: The organizations were highly selected in doing at least 5 cpgs; the organization had to have explicit policies, and many did not; they had to have systematic reviews etc. Therefore, the sampling favors the hypothesis because these are among the best organizations, trying the hardest, and therefore other organizations would only be worse. The authors could emphasize this in the discussion.

- The iom cpgs standards are not “validated”; however, there have been few if any challenges, and there is redundancy so that if several standards are not met, the ones that are met would be adequate to assure at least a reasonable level of firewalls around coi. The authors do mention this in the limitations section, and they could expand on this.

- In terms of remedies the authors might discuss why the clearinghouse or equivalent should be reestablished, because the pressure to not address coi is so great that only a centralized government or quasi government body can exert pressure on the organizations. I realize that this my perspective, but the clearinghouse during its existence seemed to be widely respected and many organizations attempted to conform to clearinghouse criteria.

Reviewer #3: The present manuscript is a great work, that updates the evidence on COIs policies of CPGs development organizations. For improving the quality of this manuscript, some suggestions were given as following:

1. Abstract: Suggested using “i) the number of” instead of “the number of i)” in line 35.

2. Punctuation seems to be missing at the end of the last sentence in the background section.

3. Suggested to amend line 64 "on the screening, diagnosis, prevention, and treatment of a wide variety of diseases" to "on the prevention, screening, diagnosis, and treatment of a wide variety of diseases" to make the expression more logical.

4. If IOM standards and IOM criteria have the same meaning, Suggested uniform expression across the text. Such as lines 91, 112, 134, 228.

5. In line 122, suggested describing web search in detail, such as adding search engine information.

6. In lines 152 to 154, only one author reviewed each CPG and its corresponding appendices for the presence of author COI disclosures. Whether it may lead to information omission?

7. Suggested providing details of Cohen's kappa calculation in the statistical methods section.

8. Table 1, Suggested that the standard a/b/c after the serial number correspond to the original standard to improve the legibility of the table.

9. Suggested to add corresponding attachments for the content corresponding to lines 148 to 154 and lines 180 to 184 so that readers can understand more details.

10. In line 234, Suggested adding the confidence interval of Cohen’s kappa.

11. If "IOM standard" in line 214 and "standard" in line 270 have the same meaning, suggested to express them uniformly.

Reviewer #4: The authors produced a relevant, useful and well-researched paper. The study investigated an important area of clinical practice guideline (CPG) development (namely, identification and management of conflicts of interest) and discussed their findings in the context of exisitng literature in a practical and fair manner. I agree with the study limitations discussed in the paper. I consider the paper suitable for publication (with minor amendments), and believe it will be a valuable addition to the current literature base. Some minor issues/points for clarification are provided below.

GENERAL COMMENTS

1. Some corrections to the grammar required in both the abstract and main text to improve clarity. For example:

- In the abstract - it is not clear if ‘previous evidence’ (line 28/29) refers to previous studies (like Norris et al) or the IOM report

- In the introduction - the definition of CPGs as ‘documents that formulate recommendations’ could be reworded to include its purpose (i.e. clinical recommendations to optimise patient care)

- In the introduction - consider rewording the statement in line 72 ‘Moreover, CPGs often do not have access to robust high-quality evidence…’ to ‘CPG developers often do not have access to robust high-quality evidence…’

2. The keywords listed in manuscript does not include ‘COI’

3. Consider adding a clear aim/objective of the study in the introduction or methods section, either in addition to the statement in line 99/100 or by rewording/expanding that statement.

METHODS SECTION

1. Suggest the authors clarify why the fact that the National Guidelines Clearinghouse is no longer available is relevant to this study (e.g. it is the database that was used by Norris et al)

2. It is unclear how/why the guideline databases were chosen for this study and what their potential limitations are.

3. Authors can include a statement on why the seven IOM standards used in the comparison were selected (e.g. COI-specific criteria used by Norris et al)

RESULTS

1. The CPG developing organisations identified (presented in Table 2) are based in the United States (n=23), Canada (n=3), Europe (n=16), Asia (1) and ‘other’ (n=3). Based on the description of the CPGs included under ‘other’ it appears that no CPG developers based in Africa or South America that met the inclusion criteria were identified. The authors may want to consider providing some thoughts on this – it may be an important finding in itself or potentially indicate a limitation of the databases searched?

2. In the Disclosure section (line 191/192), the authors state that “A significant majority of these (33, 92%) policies specified that disclosures must occur prior…” It will be helpful if the authors can explain why they considered 34% a ‘significant majority’.

3. In the discussion section, the authors state (line 308/309) they only included ‘commercial COI’ in their assessment. This is not clearly stated in the analysis presented in the results section under ‘Management of COI’. It is mentioned in the methods section that this data was extracted.

4. The strategy for managing CPG panellists’ COI described under ‘Management of COI’ (line 208-212) only describes 33 policies, while 36 organisations had COI policies. The authors may want to consider including a description of the content of the other 3 policies as well for the sake of completeness.

5. Composition of COI panels - Clarification required on two points:

- Lines 214 to 217: The authors state that 13 organisations required al chairs and co-chairs to be free of COI. However, Table 3 (last column - IOM standard 2.4d) reports the number of organisation to be 15.

- The overall analysis (management of chair and co-chair COI) only describes 32 policies (13 required all chairs and co-chairs to be free of COI, 12 had no such requirement, 7 allowed one chair or co-chair to have a COI).

6. Figure 1 – The authors could consider adding ‘IOM standards’ in the figure title or x-axis label

7. The numbers of CPG developing organisations with COI policies reported in Table 2 and Table 3 don’t align. For example, Table 3 reports 14 European CPG developing organisations with COI policies, while only 13 is reported in Table 2. In addition, Table 3 does not include Canada or Asia (as in Table 2). The first column of Table 3 suggests that all 36 CPGs with COI policies should be included in those numbers. Explanation for these discrepancies is required, e.g. if some of the fields were merged (e.g. US and Canada CPGs reported in Table 2 were combined and presented as US in Table 3) or the type of organisations included under the different fields changed (e.g. CPG organisations included under ‘other’).

DISCUSSION

1. The meaning of the second part of the following statement (line 282/283) is not clear: ‘Further improvement could be achieved through increased transparency and identification of trustworthy guidelines.’ Unclear what aspect ‘…identification of trustworthy guidelines…’ will improve. It is also unclear if the author is referring to clinical practice guidelines or COI guidelines (like the IOM standards).

2. Authors may want to clarify statement in line 315-318: ‘Although our study is focused on the extent to which published COI policies adhere to IOM guidelines and so does not provide evidence of how bias may affect CPG recommendations, limited data suggest that COI can undermine their objectivity. Thus, recommendation of a drug may be more likely when authors have a COI with the pharmaceutical company producing that drug.’ It is not clear if ‘limited data’ refers to data collected as part of this study, or by other studies (as mentioned in the next sentence of that paragraph – line 317/318).

6. PLOS authors have the option to publish the peer review history of their article (what does this mean?). If published, this will include your full peer review and any attached files.

Reviewer #1: **Yes: **Joel Lexchin

Reviewer #2: **Yes: **Sheldon Greenfield

Reviewer #3: No

Reviewer #4: No

---

## [Author Response · Author response to Decision Letter 0]

27 Feb 2021

We appreciate the opportunity to revise our manuscript. All comments have been responded to point-by-point in our attached response letter.

---

## [Editor Report · Decision Letter 1]

3 Mar 2021

PONE-D-21-02302R1

Analysis of conflict of interest policies among organizations producing clinical practice guidelines

PLOS ONE

Dear Dr. Brems,

Thank you for submitting your manuscript to PLOS ONE. After careful consideration, we feel that it has merit but does not fully meet PLOS ONE’s publication criteria as it currently stands. Therefore, we invite you to submit a revised version of the manuscript that addresses the points raised during the review process.

Thank you for your detailed response to the reviewers' comments. There are a few outstanding things I ask you to please address before we can proceed to a decision:

- the abstract conclusion is rather abrupt; would you consider adding a sentence or two on the implications of these findings for policy or practice?

- you did not address Reviewer 1's comments regarding clearly defining conflict of interest in the Introduction. The definition you provide on lines 67-71 cites the IOM, but only refers to non-financial interests, which is not the main focus of the IOM report. You then refer to "financial COI" (which reflects standard 2.3, which you assess) and then "commercial COI", but do not define these terms. In the discussion, you re-introduce these and "intellectual" COI, but state that you only assessed "financial" COI. Please address the use of multiple terminologies and clearly define these concepts up front. I would suggest using the IOM definition and then defining the concept in terms of what you will actually go on to assess. Introducing the "intellectual" COI prior to the discussion (and without defining it) is confusing.

- Reviewer 1 also referred to conflicts of interest that may arise at the organisational level. You may want to comment in the Discussion about whether/how these standards could be adapted to address situations where a guideline developer accepts funding from an industry with an interest in the guideline topic, for example.

- You comment on the generalisability of the sample and also on the geographic representation, but attribute this to exclusions based on language (which applies to South America but Africa less so). You do not include any guidelines produced by or with the WHO - might this affect your sample? You also do not include any from Australia/New Zealand (e.g. National Health and Medical Research Council). Please reflect on the strengths/limitations of your sample in terms of generalisability of findings beyond language restrictions alone.

We look forward to receiving your revised manuscript.

Kind regards,

Quinn Grundy, PhD, RN

Academic Editor

PLOS ONE
---

## [Author Response · Author response to Decision Letter 1]

13 Mar 2021

We appreciate the opportunity to revise our manuscript. All comments were quite helpful and assisted us in improving the quality of the manuscript. We have responded in detail to each comment in the attached 'Response to Reviewers' letter. We have i) added to the Conclusion of the abstract; ii) expanded on the definition of COI in the Introduction to include financial, commercial, non-commercial, intellectual COI; iii) commented on the impact and relevance of organizational COI; and iv) further expounded on the limitations of our sampling as it relates to geographic representation

---

## [Editor Report · Decision Letter 2]

16 Mar 2021

Analysis of conflict of interest policies among organizations producing clinical practice guidelines

PONE-D-21-02302R2

Dear Dr. Brems,

We’re pleased to inform you that your manuscript has been judged scientifically suitable for publication and will be formally accepted for publication once it meets all outstanding technical requirements.

Kind regards,

Quinn Grundy, PhD, RN

Academic Editor

PLOS ONE
---

## [Editor Report · Acceptance letter]

18 Mar 2021

PONE-D-21-02302R2 

Analysis of conflict of interest policies among organizations producing clinical practice guidelines 

Dear Dr. Brems:

I'm pleased to inform you that your manuscript has been deemed suitable for publication in PLOS ONE. Congratulations! Your manuscript is now with our production department. 

Kind regards, 

on behalf of

Dr. Quinn Grundy 

Academic Editor

PLOS ONE